# Comparative Gene Co-expression Network Analysis of Proviral and Antiviral Responses to Dengue Virus-2 (DENV-2) and Zika Virus (ZIKV) Infection in Human Neural Progenitor Cells (hNPCs)

**J. K. Owaresat**[1], **Diptta Dey**[1], **Md. Ahashan Habib Siam**[1]*, **Md. Ashraful Anam**[1],
**AMAM Zonaed Siddiki**[2]

**1** Department of Zoology, University of Chittagong, Chattogram, Bangladesh, **2** Department of Pathology and Parasitology, Faculty of Veterinary Medicine, Chittagong Veterinary and Animal Sciences University, Chattogram, Bangladesh

* ahashan@cu.ac.bd

## Abstract

The comparative proviral and antiviral mechanisms underlying host responses to Dengue virus (DENV-2) and Zika virus (ZIKV) in neural progenitor cells remain poorly understood. In this study, we first performed weighted gene co-expression network analysis (WGCNA) on RNA-seq data obtained from 30 hNPC samples infected with DENV-2 and ZIKV to compare the proviral and antiviral responses by identifying key virus-associated modules, hub genes, and enriched pathways. A total of 4,587 genes were grouped into 12 co-expression modules. The turquoise module (867 genes) showed a strong positive correlation with DENV-2 infection, whereas the green module (479 genes) showed a strong positive correlation with ZIKV infection ($r = 0.92$, adjusted $p < 1 \times 10^{-12}$). Hub genes, including CALR in the turquoise module and HMGCS1 in the green module, were found to be associated with viral replication. Functional enrichment analysis (GO, KEGG, and Reactome) revealed that both viruses modulated virus-specific and similar proviral and antiviral pathways. Among specific proviral pathways, DENV-2 infection was primarily enriched for TGF-β signaling, whereas ZIKV infection mainly engaged host translation regulation and nucleic acid-binding pathways. Similar proviral responses included endoplasmic reticulum (ER) stress and protein-folding mechanisms. Unique antiviral responses were also observed: DENV-2 triggered p53-mediated pathways, PI3K-Akt and Hippo signaling, NF-κB signaling, oxidative stress responses, and immunoglobulin class switching, whereas ZIKV infection enriched apoptotic and inflammatory pathways. Similar antiviral responses included autophagy, ubiquitin-mediated protein degradation, and Rho GTPase effector pathways. Insights from this network-based molecular study may inform the development of potential therapeutic strategies for neurological diseases associated with DENV-2 and ZIKV.

**Data availability statement:** All relevant data are within the manuscript and its Supporting Information files.

**Funding:** The author(s) received no specific funding for this work.

**Competing interests:** The authors have declared that no competing interests exist.

## 1. Introduction

Dengue virus (DENV) and Zika virus (ZIKV), both mosquito-borne flaviviruses, have emerged as major global public health threats in tropical and subtropical regions, and are increasingly spreading to new areas [1,2]. Both viruses are primarily transmitted by *Aedes albopictus* and *Aedes aegypti* mosquitoes. The worldwide expansion of these vectors, driven by international trade and human travel, has facilitated the emergence and re-emergence of DENV and ZIKV outbreaks [1]. Climate change is expected to further intensify this trend. Global temperatures are projected to rise by approximately 2.5–2.9 °C by the end of the century [3], potentially accelerating the range expansion of mosquitoes and enhancing the dynamics of arbovirus transmission [4]. In addition to causing significant morbidity and mortality, DENV and ZIKV infections generate considerable social and economic impacts, including extended recovery periods that reduce workforce productivity and strain public health systems [5,6]. Although DENV and ZIKV are single-stranded RNA viruses belonging to the Flaviviridae family [7], they exhibit distinct clinical manifestations and pathogenic outcomes in infected individuals. Primary DENV infections in children and adults typically result in dengue fever (DF), a disease characterized by fever and non-specific symptoms such as headache, retro-orbital pain, myalgia, and occasionally hemorrhagic manifestations [8]. In a small proportion of cases, the infection progresses to dengue hemorrhagic fever (DHF), the most severe form of the disease, marked by plasma leakage, shock, and potentially fatal multi-organ complications [8]. Additionally, dengue fever can lead to organ dysfunctions affecting the liver, heart, kidneys, lungs, and hematologic system, further complicating disease outcomes [9]. Dengue virus has traditionally been regarded as a non-neurotropic pathogen, and neurological complications are relatively uncommon in dengue infection [10]. However, DENV can also infect cells of the central nervous system (CNS), leading to neurological complications such as rare encephalitis, meningitis, encephalopathy, stroke, and Guillain-Barré syndrome [10,11]. In contrast, ZIKV has received significant attention due to its neurotropism, infecting neural progenitor cells, microglia, astrocytes, and endothelial cells [12,13]. Neural progenitor cells serve as the primary target after this virus crosses the blood-brain barrier [14]. During pregnancy, ZIKV infection can lead to congenital Zika syndrome in infants, resulting in microcephaly, epilepsy, lissencephaly, hydrocephalus, polymicrogyria, agyria, ventriculomegaly, holoprosencephaly, and brain calcifications [15,16]. Furthermore, ZIKV infection in adults and children has been associated with Guillain-Barré syndrome, neuropathy, and myelitis [17,18].

Brain development depends on the coordinated activity of human neural progenitor cells (hNPCs), which give rise to diverse neuronal and glial populations that form the central nervous system (CNS) [19,20]. Because hNPCs are particularly vulnerable to neurotropic flaviviruses, investigating their molecular responses is essential for understanding virus-induced neuropathogenesis. A previous study [21] examined host transcriptional responses, alterations in translation efficiency pathway-level responses, regulatory motifs in mRNA UTRs, and viral genome translation patterns using RNA-seq and Ribo-seq in hNPCs infected with DENV-2 (strain

16681) and ZIKV (strain IbH 30656, NR-50066). This study used hiPSC-derived hNPCs under the oversight of the Institutional Review Board (IRB) and the Embryonic Stem Cell Research Oversight (ESCRO) committees at the Icahn School of Medicine at Mount Sinai, New York, USA. However, the analysis did not investigate higher-order gene-gene interactions in complex regulatory networks to understand how groups of genes act together in response to viral infection or identify key regulatory "hub" genes controlling these networks. Gene Co-expression Network Analysis (WGCNA) addresses this gap by clustering highly correlated genes into modules, summarizing them with module eigengenes or intramodular hub genes, and linking modules to one another and to external traits [22]. This network-based analysis framework enhances the detection of low-abundance genes or genes with small changes in expression without losing critical information and has been successfully applied to identify key pathways, hub genes, and upstream regulators across multiple diseases [23–25]. Therefore, in the present study, we reused RNA-seq datasets of DENV-2- and ZIKV-infected hNPCs from the previous study [20] and applied WGCNA to compare the proviral and antiviral responses by identifying the most enriched co-expression modules, characterizing hub genes, and evaluating their functional enrichment. Understanding the molecular networks that govern the proviral and antiviral responses in hNPCs can inform the development of targeted therapeutics, vaccines, and diagnostic biomarkers to mitigate neurodevelopmental consequences associated with DENV-2 and ZIKV infections.

## 2. Materials and methods

### 2.1. Data retrieval and sample selection

Weighted gene co-expression network analysis (WGCNA) is widely used to identify coordinated gene modules and key regulatory genes that control the proviral and antiviral responses during viral infection. Although a previous study [21] characterized transcriptional and translational responses in human neural progenitor cells (hNPCs) infected with DENV-2 (strain 16681) and ZIKV (strain IbH 30656, NR-50066), it did not investigate co-expression network patterns in these infected neural cells. For our study, we retrieved publicly available RNA-seq data (BioProject PRJNA854905, GEO: GSE207347) from NCBI. Sequences were included if they met the following criteria: i) Derived from human neural progenitor cells (hNPCs) to ensure biological relevance to neurotropic flavivirus infection, ii) Infections with both DENV and ZIKV to enable direct comparative network analysis, iii) Sufficient post-infection duration across all samples to minimize temporal variability in gene expression, iv) Adequate and balanced biological replicates per condition to ensure statistical robustness for co-expression network construction, v) Raw sequencing data and complete metadata must be publicly available, vi) Virus strains must be clearly defined to ensure reproducibility and biological interpretability, vii) Only host (*Homo sapiens*) transcriptome RNA-seq data were considered to investigate host gene co-expression responses. Based on these criteria, we downloaded 30 high-quality single-end RNA-seq samples using Fastq-dump software SRA Toolkit v3.0.3 [26]. The SRA accessions and sample information are provided in S1 Table. Fifteen samples were included for each infection type, derived from two independent human donors (Clone A and Clone B), designated as batches A and B. Batch A included 8 DENV-2-infected and 8 ZIKV-infected samples, while batch B included 7 DENV-2-infected and 7 ZIKV-infected samples, resulting in a total of 30 biological replicates. According to the original study [21], the provided tissue samples were reprogrammed into human induced pluripotent stem cells (hiPSCs) under the oversight and approval of the Institutional Review Board (IRB) and the Embryonic Stem Cell Research Oversight (ESCRO) committees at the Icahn School of Medicine at Mount Sinai, New York, USA. The hiPSCs were subsequently differentiated into hNPCs, which were infected in vitro with ZIKV IbH 30656, originally isolated from a human in Ibadan, Nigeria, and DENV-2 strain 16,681, obtained from the World Arbovirus Reference Center, University of Texas Medical Branch, Galveston, TX, USA. At 72 hours post-infection, cells were collected for ribosome foot printing and mRNA sequencing to generate the RNA datasets used in this study.

## 2.2. Data preparation and prep-process of RNA-Sequencing data

FASTQ files were first evaluated for quality using FastQC v0.12.1 [27] to identify potential adapter contamination and assess overall read quality. Raw RNA-seq reads were then processed to remove adapter sequences and low-quality bases using Cutadapt v4.4 [28] and fastp v0.23.4 [29]. Cutadapt was applied with a minimum overlap of 5 bases (-O 5) to trim multiple Illumina adapter sequences. Subsequently, fastp performed sliding-window trimming with a window size of 4 and a mean quality cutoff of Q15, removing reads shorter than 15 bases and ensuring that no more than 40% of bases in a read had a quality score below 15. To further refine the dataset, reads mapping to viral RNA, human ribosomal RNA (rRNA), transfer RNA (tRNA), and microRNA (miRNA) were sequentially filtered out by alignment to the corresponding reference sequences using Bowtie2 v2.5.2 [30], and only unaligned reads were retained for downstream analysis.

## 2.3. Reads quantification

Filtered reads were aligned to the human reference genome GRCh38.p14 (GCA_000001405.29) obtained from the Ensembl database (https://www.ensembl.org) using HISAT2 (v2.2.1) [31]. The resulting SAM files were converted to BAM format and sorted with SAMtools v1.19.2 [32]. Gene-level read counts were generated using featureCounts v2.1.1 [33], which assigns reads to exonic regions based on the gene_id attribute in *Homo sapiens*. GRCh38.114. gtf annotation file from Ensembl. Lowly expressed genes were filtered out using the filterByExpr function from edgeR v4.0.16 [34], and only genes annotated as protein-coding were retained with rtracklayer v1.66.0 [35]. Variance-stabilizing transformation (VST) [36] was applied using DESeq2 (v1.46.0) [37] to normalize the data for library size and stabilize variance. Finally, batch effects were corrected using ComBat from the sva package v3.54.0 [38], adjusting for sample batch.

## 2.4. Construction of gene co-expression network

Pearson correlation coefficients within and between the 30 RNA-seq samples were calculated using pheatmap v1.0.13 [39], followed by principal component analysis (PCA) to assess differences between groups and sample reproducibility within groups, using plotPCA() and ggplot2 v3.5.2 [40]. The WGCNA package v1.73 [22] was then applied to construct unsigned, scale-free weighted gene co-expression networks and identify modules of genes with highly correlated expression across all 30 samples. Lowly expressed genes or those with excessive missing values were removed using the goodSampleaGenes function, leaving 4,587 high-quality genes for network construction. The soft-thresholding power ($\beta$) was selected using the pickSoftThreshold function over a range of 1–20, achieving a scale-free topology fit ($R^2 > 0.8$) and mean connectivity > 100. Using the selected $\beta$, an adjacency matrix was computed to quantify gene connection strengths, which was subsequently transformed into a Topological Overlap Matrix (TOM) to measure network interconnectedness [41]. Genes were clustered based on TOM dissimilarity to detect modules of co-expressed genes, and the blockwiseModules function was used to construct the network with a minimum module size of 30. Dendrograms and module colors were visualized using the plotDendroAndColors function to illustrate clustering into distinct co-expression modules. Hub genes with the highest intramodular connectivity in each module were identified using the chooseTopHubInEachModule function. For network visualization, co-expression relationships between each hub gene and the top 30 most connected genes were extracted from the TOM, but only for the modules most strongly associated with DENV-2 and ZIKV infection. These networks were visualized using the igraph package (v2.2.0) in *R* (v4.3.1) [42]. Module-trait relationships were assessed by correlating module eigengenes (MEs) with infection status for DENV-2 and ZIKV across all 30 samples, visualized as a heatmap using the Complex Heatmap package. Additionally, correlations between module membership (MM) and gene significance (GS) were calculated to identify key genes within modules most strongly associated with the infection traits Infected with DENV-2 and Infected with ZIKV.

## 2.5. Functional enrichment analysis

Functional enrichment analysis of the modules most strongly associated with DENV-2 and ZIKV infection was performed using the cluster Profiler package (v4.10.1) [43]. Gene Ontology (GO) enrichment analysis was conducted across the Biological Process (BP), Cellular Component (CC), and Molecular Function (MF) ontologies. To further characterize the biological functions of these key modules, KEGG pathway enrichment and Reactome pathway analyses were also performed. Statistical significance was set at an adjusted p-value < 0.05 using the Benjamini-Hochberg (BH) method to control for multiple testing. All code and the two final datasets used in this analysis are provided in the Supplementary Information (S1 File, S1 Data, and S2 Data).

## 3. Results

### 3.1. Gene co-expression network analysis

In this study, we applied weighted gene co-expression network analysis (WGCNA) to examine the comparative proviral and antiviral responses in DENV-2- and ZIKV-infected human neural progenitor cells. Principal component analysis (PCA) of the 500 most variable genes from the variance-stabilized (VST) expression matrix showed clear separation between DENV-2 and ZIKV samples, reflecting strong pathogen-specific biological variance (S1 Fig). The optimal soft-thresholding power ($\beta$) for constructing an unsigned scale-free network was determined based on the WGCNA algorithm. A value of $\beta = 10$ was selected, achieving a scale-free topology fit index greater than 0.85 while maintaining a mean connectivity of 12 (S2 Fig). Using this parameter, 4,587 filtered genes were clustered into 12 co-expression modules, including the grey module. The grey module contained 129 unassigned genes that could not be incorporated into any module and were excluded from further analysis (Fig 1a). Among the remaining modules, turquoise (867 genes), blue (682 genes), and brown (667 genes) were the largest, whereas green-yellow was the smallest, containing 33 genes (Fig 1b).

### 3.2. Module-trait correlation analysis and module validation

Correlation analysis between module eigengenes and infection status identified two modules with the strongest associations: the turquoise module was highly positively correlated with DENV-2 infection (Pearson $r = 0.92$, adjusted $P = 2.9 \times 10^{-13}$), while the green module was strongly associated with ZIKV infection ($r = 0.92$, adjusted $P = 7.2 \times 10^{-13}$) (Fig 2). Positive and negative correlations were visualized in red and blue, respectively, with darker shades representing stronger associations. These results indicate that the identified modules capture biologically meaningful changes in gene expression specific to each viral infection. Module robustness was further confirmed by evaluating gene significance (GS) and module membership (MM) values within each module. Highly positive and significant correlations were observed (correlation range = 0.88–0.91), with genes showing the highest GS-MM correlations considered hub genes within their respective modules (Fig 3).

### 3.3. Hub gene identification

Notably, the analysis revealed key hub genes, defined as the most highly connected genes within a module that may act as central regulators of infection-specific responses. In the turquoise module associated with DENV-2 infection, the top hub gene was CALR (ENSG00000179218), which encodes the endoplasmic reticulum chaperone protein Calreticulin (Table 1, Fig 4a). For the green module associated with ZIKV infection, the top hub gene was HMGCS1 (ENSG00000112972), which encodes 3-hydroxy-3-methylglutaryl-CoA synthase 1, a key enzyme in the mevalonate pathway (Table 1, Fig 4b).

### 3.4. Functional enrichment analysis of turquoise and green modules

Significantly enriched Gene Ontology (GO) terms in S2 Table (a, b), KEGG pathways in S3 Table (a, b), and Reactome pathways in S4 Table (a, b) (Benjamini–Hochberg adjusted p < 0.05) were identified for the turquoise module (associated

a

### Gene Cluster Dendrogram

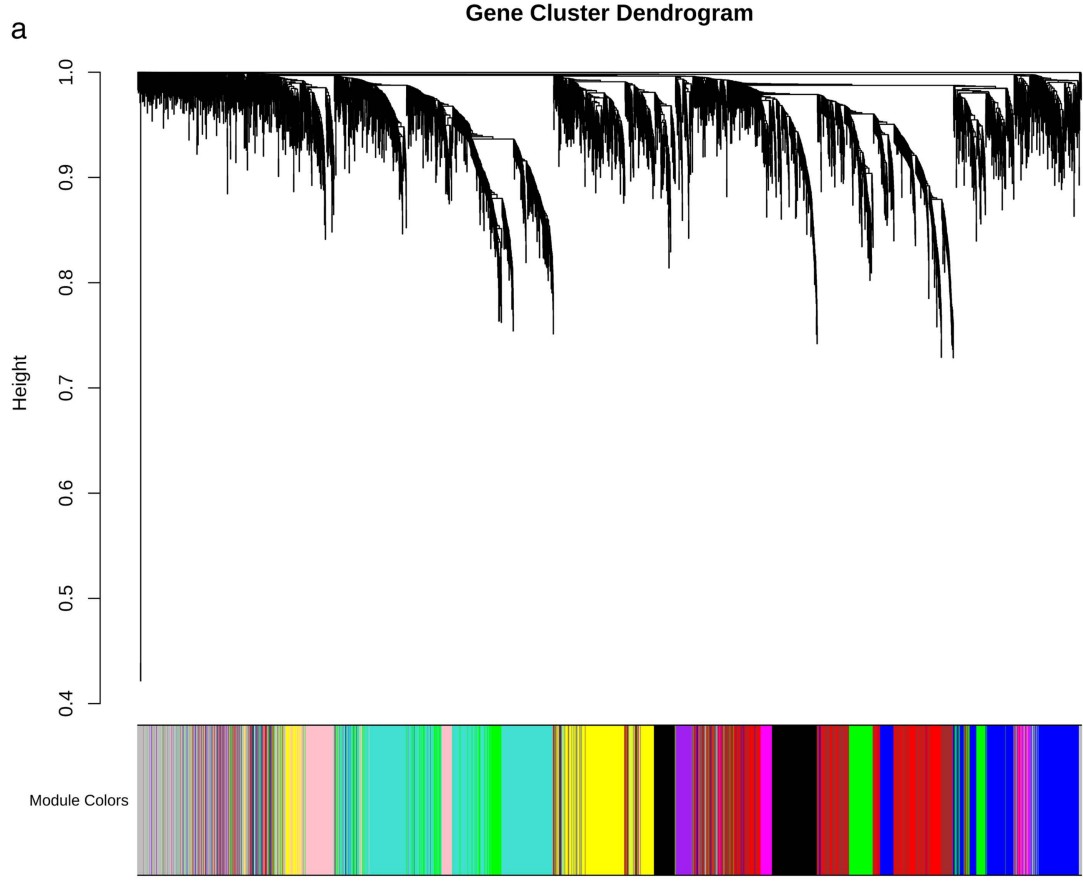

b

## Gene Count per Module

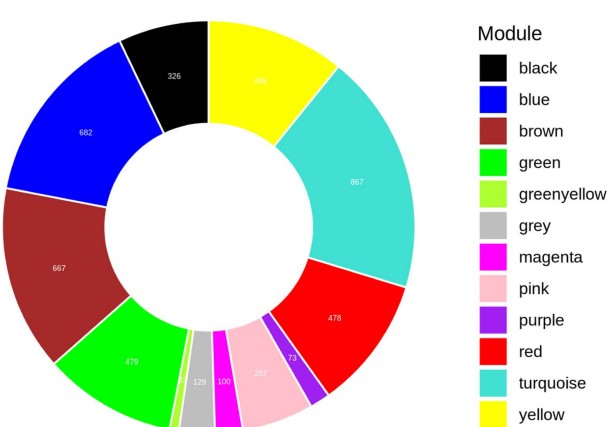

**Fig 1. (a) Hierarchical clustering of gene expression across 30 combined DENV-2- and ZIKV-infected hNPC RNA samples.** Genes were organized into 12 distinct modules, including the grey module. On the X-axis, each color corresponds to a specific gene module, while the Y-axis represents expression similarity. The dynamic tree cut function in WGCNA was applied to prune the dendrogram at various heights based on these similarities. **(b)** Distribution of genes within each module, shown according to their respective module colors (including the grey module).

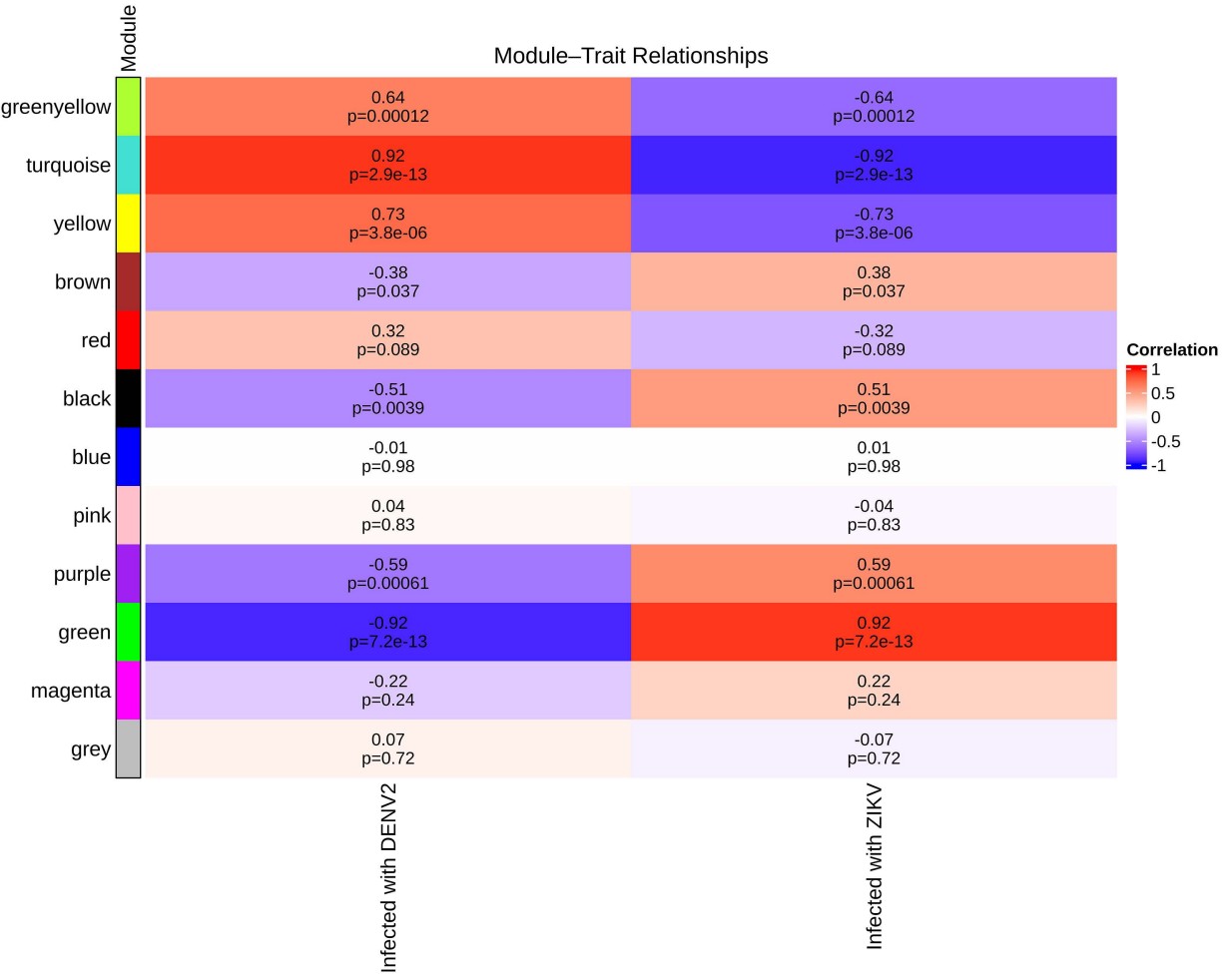

**Fig 2. Heatmap depicting the correlations between gene co-expression modules and infection stages or traits.** Rows correspond to individual modules, and columns represent different stages of infection. Strong positive correlations are shown in dark red (r near 1), while strong negative correlations are shown in dark blue (r near –1). Correlations were considered significant if Pearson's r exceeded 0.8 with P<0.05.

with DENV-2 infection) and the green module (associated with ZIKV infection). The turquoise module showed enrichment for 600 significant GO terms, including 407 Biological Process (BP), 138 Cellular Component (CC), and 55 Molecular Function (MF) terms in DENV-2 infection. In contrast, the green module associated with ZIKV infection contained 230 significant GO terms, comprising 150 BP, 61 CC, and 19 MF categories. Although the top-ranked GO, KEGG, and Reactome terms in both infection conditions were largely related to fundamental cellular processes, our analysis focused specifically on antiviral and immune-related responses relevant to viral infection. Within the turquoise module for DENV-2 infection (Fig 5a), BP enrichment highlighted strong involvement of transforming growth factor beta (TGF-β) signaling and its regulation (#12, #15, #36, #70, #76, #133, #143), endoplasmic reticulum (ER) stress adaptation and autophagy (#42, #58, #62), small GTPase and p53-mediated signaling pathways (#72, #117, #120, #124), hydrogen peroxide metabolism and NF-κB activation (#150, #196, #335), as well as immunoglobulin isotype switching and diversification (#236, #332, #396, #408). At the CC level, enriched antiviral and immune-associated structures included tight junctions (#478), cytoplasmic stress granules (#482), membrane rafts (#483), proton-transporting two-sector ATPase complexes (#495), autophagic

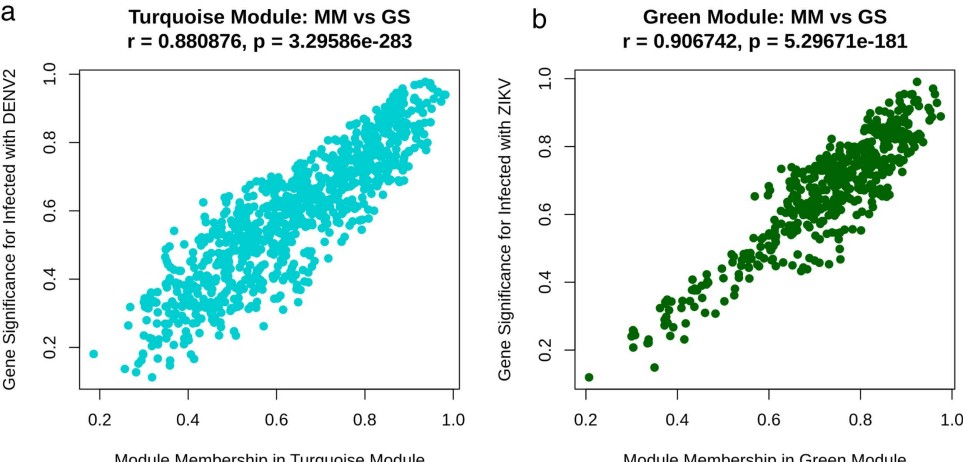

Fig 3. Scatter plots showing the relationship between gene significance (GS) and module membership (MM) for modules associated with DENV-2 and ZIKV infections. Strong positive correlations were observed in both modules: (a) the turquoise module for DENV-2 infection (r = 0.88, P = 3.296 × 10⁻²⁸³), and (b) the green module for ZIKV infection (r = 0.91, P = 5.297 × 10⁻¹⁸¹). Correlations were considered significant if Pearson's r exceeded 0.8 with P < 0.05.

Table 1. Hub genes identified in 11 modules (excluding the green module) along with their corresponding encoded proteins.

| Module color | Number of genes | Hub gene | Protein encoded by hub genes |
|---|---|---|---|
| Yellow | 496 | AVL9 | AVL9 cell migration associated |
| Green | 479 | HMGCS1 | 3-hydroxy-3-methylglutaryl-CoA synthase 1 |
| Red | 478 | WDFY2 | WD repeat and FYVE domain containing 2 |
| Blue | 682 | FER | FER tyrosine kinase |
| Green yellow | 33 | WDR19 | WD repeat domain 19 |
| Purple | 73 | DRAXIN | dorsal inhibitory axon guidance protein |
| Pink | 257 | GPRC5B | G protein-coupled receptor class C group 5 member B |
| Magenta | 100 | SCG2 | secretogranin II |
| Turquoise | 867 | CALR | calreticulin |
| Brown | 667 | CAPZA2 | capping actin protein of muscle Z-line subunit alpha 2 |
| Black | 326 | UBA52 | ubiquitin A-52 residue ribosomal protein fusion product 1 |

vesicles (#503, #515), adherens junctions (#510), ubiquitin ligase complexes (#534), and the ER chaperone complex (#539). MF analysis revealed enrichment of functions related to heat shock protein binding (#560), guanyl ribonucleotide binding (#573), ubiquitin ligase binding (#574, #581), ATPase binding (#587), unfolded protein binding (#600), and protein kinase regulator activity (#601). KEGG pathway analysis (Fig 5b) identified enrichment in pathways associated with viral and immune responses, including COVID-19 (#06), prion disease (#10), systemic lupus erythematosus (#13), Hippo signaling pathway (#18), neutrophil extracellular trap formation (#20), human papillomavirus infection (#23), endocytosis (#24), viral carcinogenesis (#30), phagosome (#32), autophagy (animal) (#37), and the PI3K–Akt signaling pathway (#42). Reactome pathway enrichment (Fig 5c) further supported the involvement of antiviral and immune mechanisms, highlighting pathways such as influenza infection (#2), viral mRNA translation (#13), influenza viral RNA transcription and replication (#17), RHO GTPase signaling (#10, #23, #29), SARS-CoV infections (#28, #87, #104), oxidative stress-induced senescence (#31), TGF-β signaling (#57, #127, #136), apoptosis-related DNA fragmentation (#80), MHC class II antigen presentation (#89), programmed cell death pathways (#91), response of Mycobacterium tuberculosis to phagocytosis

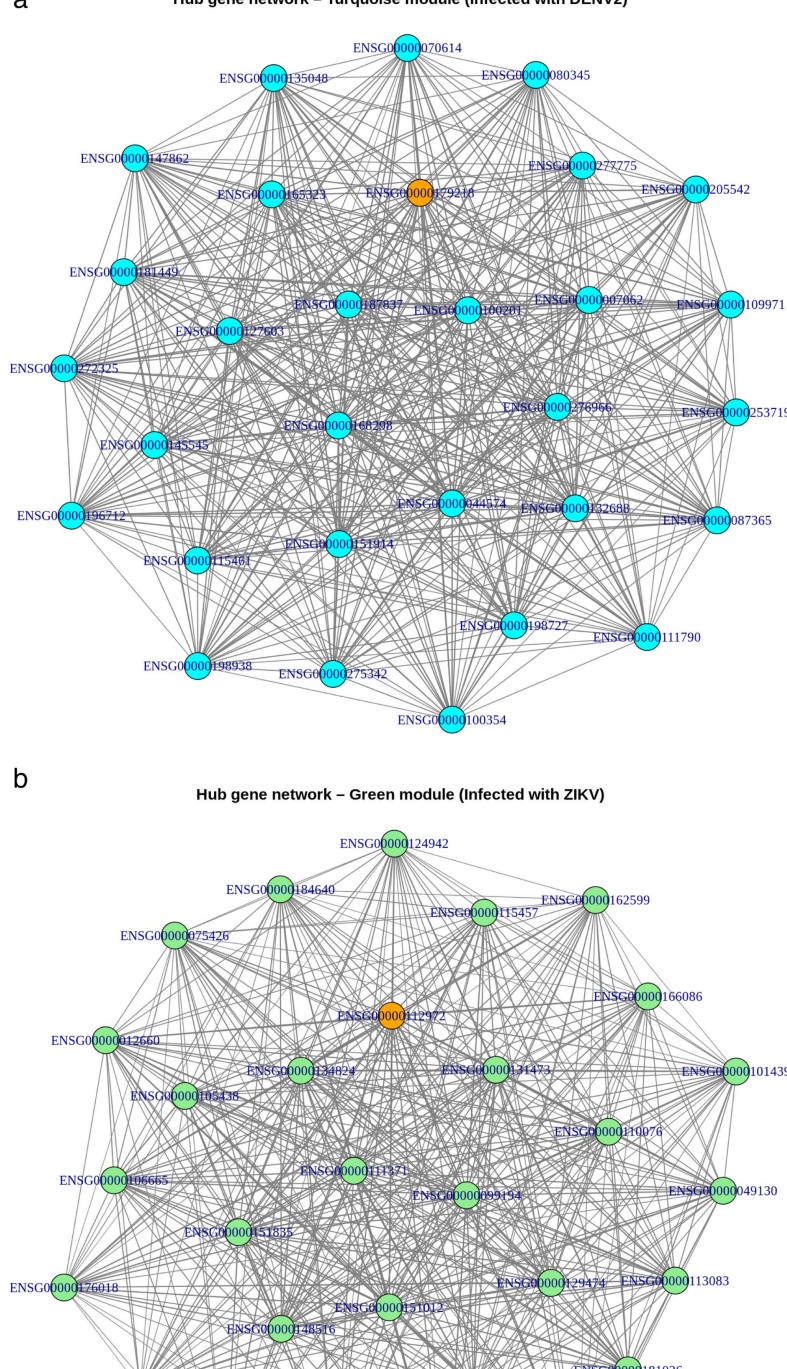

**a** Hub gene network – Turquoise module (Infected with DENV2)

**b** Hub gene network – Green module (Infected with ZIKV)

**Fig 4. Gene co-expression network plots for (a) the turquoise module linked to DENV-2 infection and (b) the green module linked to ZIKV infection, depicting the hub genes CALR (ENSG00000179218) and HMGCS1 (ENSG00000112972) together with their 30 most strongly connected genes in each module.** The hub gene is marked in bright orange. Node size corresponds to gene connectivity, while edge thickness indicates the strength of co-expression.

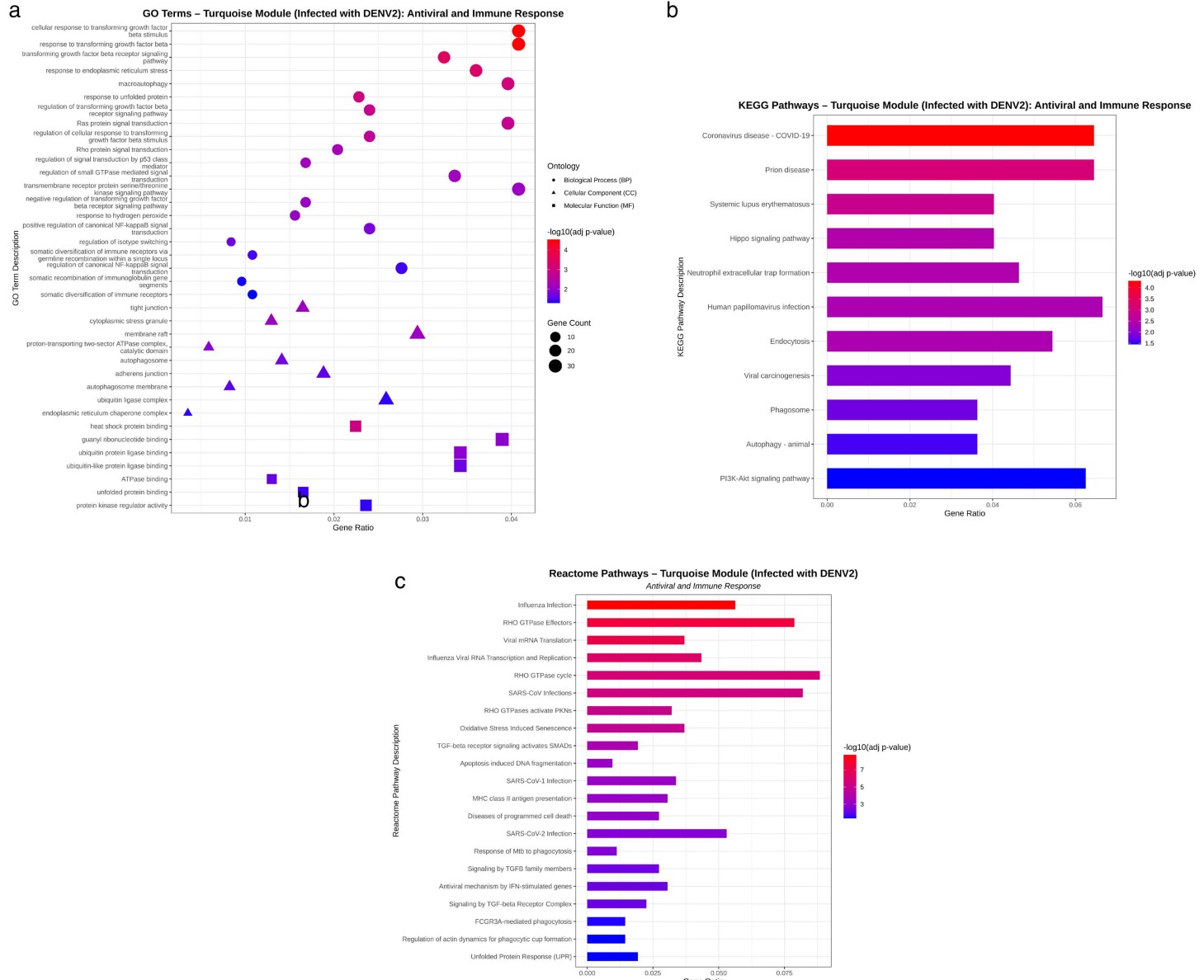

**Fig 5. (a) Bubble plot illustrating antiviral- and immune-related enriched GO terms, (b) bar plot showing enriched KEGG pathways, and (c) bar plot displaying enriched Reactome pathways for the turquoise module (DENV-2 infection).** The X-axis represents the Gene Ratio, defined as the proportion of genes annotated to a specific term relative to the total number of genes associated with that term. Bubble or bar size corresponds to the number of enriched genes, with larger sizes indicating greater gene counts. Color intensity reflects the adjusted P-value of enrichment, where red denotes higher statistical significance. All displayed GO, KEGG, and Reactome terms (FDR-adjusted P < 0.05) are ordered by increasing adjusted P-value.

(#107), interferon-stimulated genes (#132), FCGR3A-mediated phagocytosis (#195), regulation of actin dynamics during phagocytic cup formation (#203), and the unfolded protein response (UPR) (#208).

In contrast, the green module associated with ZIKV infection (Fig 6a) displayed distinct functional enrichment patterns. Biological Process (BP) analysis highlighted pathways related to viral replication and viral life cycle regulation (#15, #24,

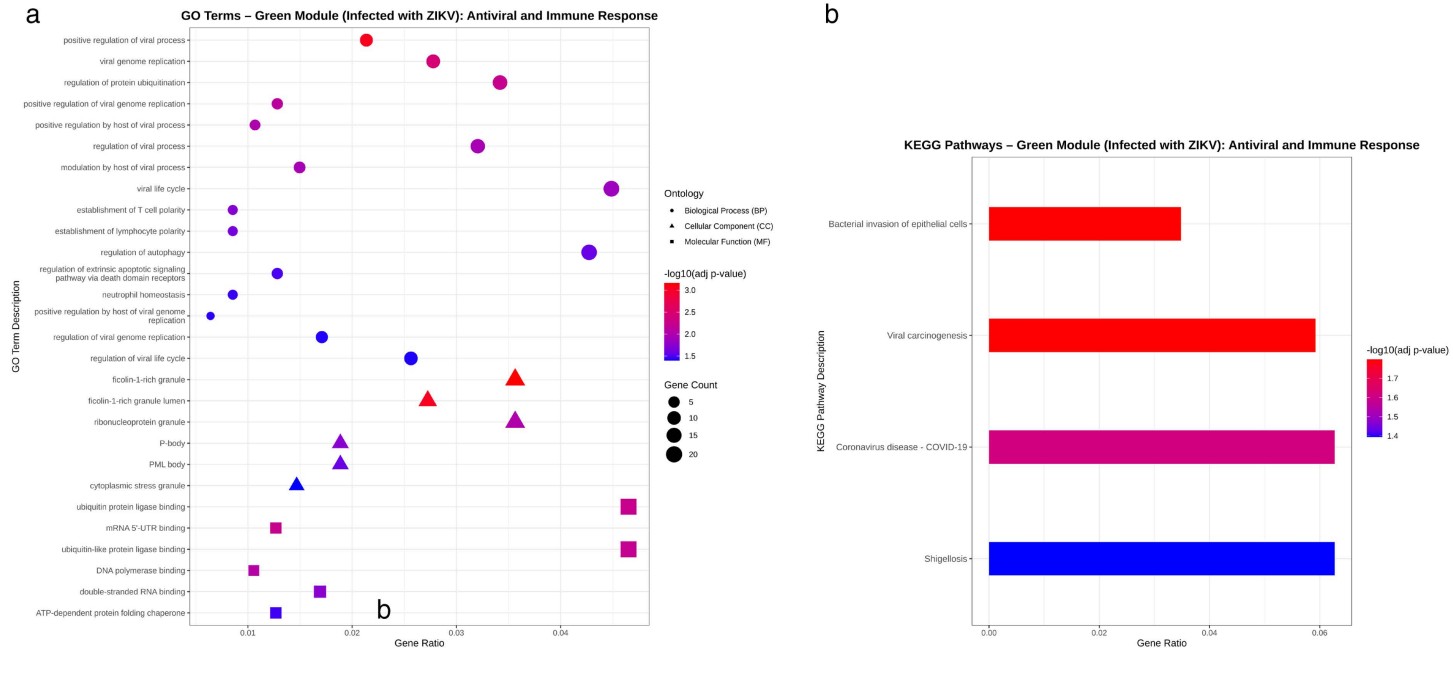

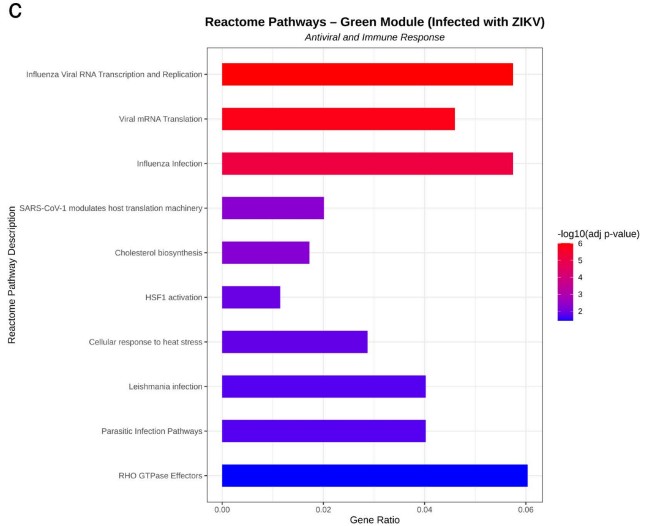

**Fig 6. (a) Bubble plot illustrating antiviral- and immune-related enriched GO terms, (b) bar plot showing enriched KEGG pathways, and (c) bar plot displaying enriched Reactome pathways for the green module (Infected with ZIKV).** The X-axis represents the Gene Ratio, defined as the proportion of genes annotated to a specific term relative to the total number of genes associated with that term. Bubble or bar size corresponds to the number of enriched genes, with larger sizes indicating greater gene counts. Color intensity reflects the adjusted P-value of enrichment, where red denotes higher statistical significance. All displayed GO, KEGG, and Reactome terms (FDR-adjusted P < 0.05) are ordered by increasing adjusted P-value.

#35, #54, #66, #135, #137), host modulation of viral infection (#43, #59, #131), the ubiquitination–autophagy axis in antiviral defense (#29, #102), and apoptotic and neutrophil-associated regulatory processes (#77, #82, #115, #123). At the Cellular Component (CC) level, enriched structures included ficolin-1–containing granules (#160, #162), ribonucleoprotein granules (#172), processing bodies (P-bodies) (#186), promyelocytic leukemia (PML) bodies (#194), and cytoplasmic

stress granules (#205), all of which are implicated in antiviral and RNA regulatory mechanisms. Molecular Function (MF) analysis further revealed enrichment in ubiquitin protein ligase binding (#215), mRNA 5′-UTR binding (#216), ubiquitin-like protein ligase binding (#219), DNA polymerase binding (#221), double-stranded RNA binding (#223), and ATP-dependent protein folding chaperone activity (#228), indicating strong involvement of RNA sensing and protein quality control pathways. Consistent with the GO results, KEGG pathway analysis (Fig 6b) identified enrichment in pathways such as bacterial invasion of epithelial cells (#3), viral carcinogenesis (#4), Coronavirus disease (COVID-19) (#5), and Shigellosis (#6). Reactome pathway analysis (Fig 6c) further supported these findings, highlighting Influenza viral RNA transcription and replication (#11), viral mRNA translation (#14), influenza infection (#17), SARS-CoV-1 modulation of host translation machinery (#30), cholesterol biosynthesis (#32), HSF1 activation (#37), cellular response to heat stress (#38), Leishmania infection (#40), parasitic infection pathways (#41), and RHO GTPase effectors (#47).

## 4. Discussion

Dengue virus (DENV) and Zika virus (ZIKV) are emerging global pathogens capable of infecting human neural cells and inducing severe neurological complications such as encephalitis, microcephaly, and neurodevelopmental disorders [10–13]. Understanding how host cellular pathways of neural cells respond to these viruses at molecular level is essential for developing effective strategies to prevent and treat virus-induced neurological diseases. In this study, we aimed to compare the proviral and antiviral host responses in human neural progenitor cells (hNPCs) infected with DENV-2 and ZIKV by identifying key hub genes and strongly associated co-expression modules. Using weighted gene co-expression network analysis (WGCNA) on a previously published RNA-seq dataset containing 30 samples (15 DENV-2-infected and 15 ZIKV-infected) [21], we identified two infection-associated modules with strong correlations (r = 0.92; GS-MM > 0.80): the turquoise module associated with DENV-2 infection and the green module associated with ZIKV infection.

Functional enrichment analysis (BP, CC, MF, KEGG, and Reactome) of these two modules revealed that the two viruses interact with infected host hNPCs through both distinct and similar proviral and antiviral pathways. As a proviral strategy, both viruses suppressed host immune pathways to facilitate their replication and survival in infected human neural progenitor cells (hNPCs through different signaling pathways. The turquoise module associated with DENV-2 infection was strongly enriched for pathways related to TGF-β signaling, including important genes such as TGFB1 and TGFBR1. TGFB1 (Transforming Growth Factor-β1) acts as an immunosuppressive cytokine by inhibiting T-cell proliferation through activation of the Smad2 signaling pathway via TGFBR1 [44,45]. Previous studies have shown that the TGF-β1/Smad2 axis may support dengue virus immune evasion and promote viral replication in human cells [46]. In contrast, the green module, primarily associated with ZIKV infection, was enriched for pathways involved in viral RNA replication, viral translation, nucleic acid binding, and modulation of the host translation machinery, with several significant genes, such as ADAR and DDX3X. ADAR (Adenosine Deaminase Acting on RNA) has been shown to regulate viral genome replication by suppressing PKR activation and interferon (IFN) production in human lung epithelial and embryonic kidney cells [47], whereas DDX3X (DEAD-Box Helicase 3, X-Linked) facilitates Zika virus replication by binding and unwinding the 5′ viral RNA in human cells [48].

Both viruses also exploited endoplasmic reticulum (ER) stress and protein-folding pathways to support their life cycles in infected human neural progenitor cells (hNPCs), with similar proviral signaling mechanisms observed in both infections. Enrichment of pathways related to heat shock protein binding, chaperone activity, and the unfolded protein response (UPR) was observed in both modules. The DENV-associated turquoise module contained the ER chaperone HSPA5 (Heat Shock Protein Family A [Hsp70] Member 5), whereas the ZIKV-associated green module included HSPA4 (Heat Shock Protein Family A [Hsp110] Member 4). These molecular chaperones, activated by the UPR, assist in the folding of viral proteins under ER stress conditions, a process that is frequently hijacked by viruses to support viral protein synthesis [49,50].

Despite these proviral mechanisms, DENV-2 and ZIKV triggered antiviral responses in infected human neural progenitor cells (hNPCs). Both viruses induced antiviral processes through autophagy and ubiquitin-mediated protein

degradation, and these antiviral mechanisms were similar in both infections. In the DENV-associated module, including UBE2K, ATG2B, RB1CC1, and UVRAG were enriched, whereas the ZIKV-associated module included UBE3A and PIK3C3. UBE2K (Ubiquitin-Conjugating Enzyme E2K) and UBE3A (Ubiquitin Protein Ligase E3A) are components of the ubiquitination machinery that tag viral proteins for degradation [51,52]. ATG2B (Autophagy Related 2B), RB1CC1 (RB1 Inducible Coiled-Coil 1), UVRAG (UV Radiation Resistance Associated), and PIK3C3 (Phosphatidylinositol 3-Kinase Catalytic Subunit Type 3) contribute to autophagosome formation during viral infection [53–56]. Through autophagy, infected host cells can degrade and remove viral proteins tagged by ubiquitination, a mechanism previously observed in infections with Herpes Simplex Virus type 1 (HSV-1) and Vesicular Stomatitis Virus (VSV) [57,58]. However, Dengue and Zika virus can hijack these autophagic structures to enhance their genome replication and pathogenicity in human cells [59,60].

Several cellular signaling pathways also played important roles in regulating the viral life cycle in infected hNPCs. The DENV-associated module showed enrichment for small GTPase signaling, p53-mediated pathways, and PI3K-Akt and Hippo signaling pathways, involving genes such as RAC1 and ATM. RAC1 (Rac Family Small GTPase 1) maintains endothelial barrier integrity through regulating actin cytoskeleton remodeling and intercellular junction [61]. Interestingly, the dengue virus has been shown to transiently suppress RAC1 to facilitate viral entry and later reactivates it via the E protein to promote viral release in human endothelial cells (EAhy926) [62]. Similarly, ATM (ATM Serine/Threonine Kinase) activates p53-mediated antiviral responses and interferon-stimulated genes that restrict dengue virus replication in human liver cancer cells (HepG2) [63]. The ZIKV-associated module also showed enrichment of Rho GTPase effector pathways, including the gene DIAPH1. DIAPH1 (Diaphanous-related formin 1) acts as a host restriction factor that inhibits viral replication by regulating the cytoskeleton, specifically through Rho GTPase signaling pathways in Porcine Kidney-15 cells (PK-15 cells) [64].

In addition to intracellular signaling pathways, DENV-2 triggered several immune-related pathways. Oxidative stress responses, interferon-stimulated gene activation, apoptosis-associated DNA fragmentation, programmed cell death pathways, and NF-κB signaling were enriched in DENV-2-associated module, including key regulators such as TNFR1/TNFRSF1A and NFE2L2. Previous studies have shown that activation of TNFR1/TNFRSF1A recruits the RIPK1 complex, leading to NF-κB–mediated antiviral responses that promote cell survival through induction of anti-apoptotic genes and proinflammatory cytokines in infected human hepatoma 7 cells (Huh7) and human embryonic kidney 293T cells (HEK293T) [65]. Additionally, elevated $H_2O_2$-induced oxidative stress can further stimulate NF-κB signaling [66]. In contrast, NFE2L2 (Nuclear Factor, Erythroid 2-Like 2) reduces oxidative stress, which may attenuate innate immune responses and consequently facilitate viral replication in K562 megakaryocytes [67]. Additional antiviral immune responses were identified in the DENV-2-associated turquoise module, including pathways related to immunoglobulin class switching, neutrophil extracellular trap formation, antigen presentation, and phagocytosis, involving genes such as RIF1. RIF1 (Replication Timing Regulatory Factor 1) is a multifunctional protein that protects DNA ends during class switch recombination and regulates late-stage B-cell differentiation in mice [68]. For ZIKV infection, the green module was enriched for apoptotic and inflammatory pathways, involving genes such as DDX3X and HMGB1. Functionally, DDX3X has been shown to promote pyroptosis by activating the NLRP3 inflammasome [69,70]. However, in human A549 epithelial cells, ZIKV can delay both intrinsic and extrinsic apoptosis by modulating Bcl-2 family proteins, thereby supporting viral persistence [71]. Additionally, extracellular HMGB1 amplifies inflammatory responses by stimulating the release of cytokines that promote inflammation from macrophages and neutrophils [72], as reported in Dengue and West Nile virus infections [73,74], and similarly observed in Zika-infected Huh7 liver cells [75].

Hub gene analysis further highlighted distinct central regulators for each viral infection. The turquoise module identified CALR (calreticulin) as a key hub gene in DENV-2-infected hNPCs. Dengue virus infection leads to the accumulation of unfolded and misfolded proteins, triggering endoplasmic reticulum (ER) stress [76]. CALR contributes to the activation of the unfolded protein response (UPR) [77,78] and facilitates proper folding and assembly of viral proteins [79], promoting cell survival and viral replication. In contrast, the green module associated with ZIKV infection identified HMGCS1 (3-hydroxy-3-methylglutaryl-CoA synthase 1) as the central hub gene. HMGCS1 regulates the mevalonate pathway, which

controls cholesterol biosynthesis [79]. This pathway is utilized by ZIKV to form replication complexes on ER-derived membranes, facilitating viral RNA synthesis and replication in human hepatoma Huh7 and Madin-Darby Canine Kidney (MDCK cells) [80,81].

The identification of these proviral and antiviral pathways in human neural progenitor cells (hNPCs) provides important insight into the molecular mechanisms underlying virus-induced neurological diseases. Disruption of essential cellular signaling pathways can significantly impair the normal survival, growth, and differentiation of healthy cells. The activation of multiple antiviral and immune-regulatory pathways in DENV-2-infected neural cells may explain why neurological complications such as encephalitis, meningitis, encephalopathy, stroke, and Guillain-Barré syndrome are relatively rare in dengue infection [10,11]. In contrast, the predominance of proviral replication mechanisms in ZIKV-infected human neural progenitor cells (hNPCs) may underlie the strong neurotropic capacity of Zika virus [12–13]. This pronounced neurotropism is associated with a wide range of neurological abnormalities, including microcephaly, epilepsy, lissencephaly, hydrocephalus, polymicrogyria, agyria, ventriculomegaly, holoprosencephaly, and brain calcifications, as well as neurological disorders such as Guillain-Barré syndrome, neuropathy, and myelitis in affected individuals [14–18].

## 5. Conclusion

Our study performed WGCNA-based comparative transcriptomic analysis on DENV-2- and ZIKV-infected human neural progenitor cells (hNPCs), which were prepared under the oversight of the Institutional Review Board (IRB) and the Embryonic Stem Cell Research Oversight (ESCRO) committees at the Icahn School of Medicine at Mount Sinai, New York, USA. Results showed that both infections triggered similar as well as virus-specific proviral and antiviral pathways in human neural progenitor cells (hNPCs). Notably, key hub genes, including CALR in DENV-2 infection and HMGCS1 in ZIKV infection, highlighted virus-specific regulatory pathways that may contribute to viral replication and modulate host cellular responses in hNPCs. Given the critical role of hNPCs in brain development, dysregulation of these pathways may underlie the neurological complications associated with flavivirus infections. These comparative insights enhance our understanding of flavivirus–host interactions and may facilitate the identification of potential therapeutic targets for preventing virus-induced neurological complications. Considering the severe dengue outbreaks in Bangladesh since 2023, similar studies are urgently needed to further elucidate virus-host interactions and to guide strategies aimed at preventing flavivirus-associated neurological complications in the region.

## Supporting information

**S1 Table. SRA accessions of RNA.**
(XLSX)

**S1 Fig. Principal component analysis (PCA) of the 500 most variable genes.**
(DOCX)

**S2 Fig. The optimal soft-thresholding power (β) for constructing an unsigned scale-free network.**
(DOCX)

**S2 Table. (a) GO-Turquoise module (DENV-2); (b) GO-Green module (ZIKV).**
(XLSX)

**S3 Table. (a) KEGG Turquoise module (DENV2); (b) KEGG Green module (ZIKV).**
(XLSX)

**S4 Table. (a) Reactome Turquoise module (DENV2); (b) Reactome Green module (ZIKV).**
(XLSX)

**S1 File. All author-generated code.**
(DOCX)

**S1 Data. Metadata.**
(CSV)

**S2 Data. VST ComBat Corrected.**
(CSV)

## Author contributions

**Conceptualization:** J. K. Owaresat, M. A. Habib Siam, AMAM Zonaed Siddiki.

**Data curation:** J. K. Owaresat, D. Dey, M. A. Habib Siam, M. A. Anam.

**Formal analysis:** J. K. Owaresat, M. A. Habib Siam.

**Investigation:** J. K. Owaresat, D. Dey.

**Methodology:** J. K. Owaresat, M. A. Habib Siam, M. A. Anam.

**Supervision:** M. A. Habib Siam, AMAM Zonaed Siddiki.

**Writing – original draft:** J. K. Owaresat, M. A. Habib Siam.

**Writing – review & editing:** J. K. Owaresat, D. Dey, M. A. Anam, AMAM Zonaed Siddiki.

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
