## [Decision Letter · Decision Letter 0]

16 Feb 2026

PONE-D-25-55299Comparative Gene Co-Expression Network Analysis of Antiviral and Immune Responses to Dengue-2 and Zika Virus Infection in Human Neural Progenitor Cells ((hNPCs)PLOS One

Dear Dr. Siam,

Thank you for submitting your manuscript to PLOS ONE. After careful consideration, we feel that it has merit but does not fully meet PLOS ONE’s publication criteria as it currently stands. Therefore, we invite you to submit a revised version of the manuscript that addresses the points raised by both reviewers #1 and #2.

We look forward to receiving your revised manuscript.

Kind regards,

Qiang Shawn Chen, Ph.D.

Academic Editor

PLOS One

Journal Requirements:

Gene coexpression network during ontogeny in the yellow fever mosquito, Aedes aegypti - https://doi.org/10.1186/s12864-023-09403-4

Transcriptome analysis of Aedes aegypti Aag2 cells in response to dengue virus-2 infection - https://doi.org/10.1186/s13071-020-04294-w

(Among others)

In your revision ensure you cite all your sources (including your own works), and quote or rephrase any duplicated text outside the methods section. Further consideration is dependent on these concerns being addressed.

3. Please note that PLOS One has specific guidelines on code sharing for submissions in which author-generated code underpins the findings in the manuscript. In these cases, we expect all author-generated code to be made available without restrictions upon publication of the work. Please review our guidelines at https://journals.plos.org/plosone/s/materials-and-software-sharing#loc-sharing-code and ensure that your code is shared in a way that follows best practice and facilitates reproducibility and reuse.

4. We note that your Data Availability Statement is currently as follows: “All relevant data are within the manuscript and its Supporting Information files.”

5. We notice that your supplementary figures are uploaded with the file type 'Figure'. Please amend the file type to 'Supporting Information'. Please ensure that each Supporting Information file has a legend listed in the manuscript after the references list.

Reviewer's Responses to Questions

**Comments to the Author**

1. Is the manuscript technically sound, and do the data support the conclusions?

Reviewer #1: Yes

Reviewer #2: Yes

2. Has the statistical analysis been performed appropriately and rigorously? 

Reviewer #1: Yes

Reviewer #2: Yes

3. Have the authors made all data underlying the findings in their manuscript fully available?

Reviewer #1: Yes

Reviewer #2: Yes

4. Is the manuscript presented in an intelligible fashion and written in standard English?

Reviewer #1: Yes

Reviewer #2: Yes

5. Review Comments to the Author

Reviewer #1: This manuscript describes the in silico analysis of RNA-Seq data obtained from dengue virus and Zika virus infections in human progenitor neural cells (hNPCs). Using Weighted Comparative Gene Co-expression Network Analysis (WGCNA), the authors identified clear differences in the gene expression patterns of dengue virus versus Zika virus-infected hNPCs. While several of the activated pathways are common to both types of infections, as expected, Dengue virus activated genes module showed more of an antiviral innate immunity activation signature, while the Zika virus module showed pathways more related to nucleic acid metabolism and cellular biological processes. The manuscript is well-written (albeit figure legends are misplaced along the text; they shall be all toghter at the end after the reference list), and the results are of interest since they highlights differences in the virus-cell interactions between these two viruses of public health importance.

Comments:

1. Material and Methods section. The authors should explain better the criteria used to select the RNA-Seq samples to be downloaded and analyzed from the NCBI SRI files. It will help the reader if the authors provide details of the infection conditions used to generate the RNA-Seq data chosen for their analysis.

2. Discussion section. Authors may discuss how the results may explain the neurotropism shown by the Zika virus but not by the dengue virus in infected patients.

Reviewer #2: In the manuscript by Owaresat et al., the author perform in in-depth analysis of RNA-seq data from human cells infected with DENV or ZIKV. There analysis is described well in the methods, and their description of the results is easy to follow. The discussion of the data is interesting to the reader.

My only minor comment is regarding the data sets. Are these published anywhere else, or is this manuscript the first analysis of these data? I could not find any reference to the data except the link to the SRA reads. The authors need to clarify.

6. PLOS authors have the option to publish the peer review history of their article (what does this mean? ). If published, this will include your full peer review and any attached files.

**Do you want your identity to be public for this peer review?** For information about this choice, including consent withdrawal, please see our Privacy Policy .

Reviewer #1: No

Reviewer #2: No

---

## [Author Response · Author response to Decision Letter 1]

14 Mar 2026

Response to Reviewer’s comments

Reviewer 1

i) Method

We have revised the Materials and Methods section to clarify the criteria used for RNA-seq dataset selection. The dataset was selected based on biological relevance (human neural progenitor cells), inclusion of both DENV-2 and ZIKV infections under comparable conditions, consistent infection time point (72 h post infection), balanced sample representation, and sufficient sample size to support robust WGCNA analysis. Additional details regarding infection conditions and period have now been included in the S1 Table: SRA accessions of RNA sequences.

ii) Discussion

In the revised manuscript, we have added a paragraph in the Discussion section addressing the neurotropism of Zika virus that is marked as yellow in color in the revised manuscript.

Reviewer 2

We clarify that the RNA-seq data (BioProject PRJNA854905, GEO: GSE207347) were previously generated and published by Singh K, Martinez MG, Lin J, Gregory J, Nguyen TU, Abdelaal R, Kang K, Brennand K, Grünweller A, Ouyang Z, Phatnani H. Transcriptional and translational dynamics of Zika and dengue virus infection. Viruses. 2022 Jun 28; 14(7):1418. DOI: https://doi.org/10.3390/] that is cited as [21] in the introduction part of this revised manuscript. The author conducted transcriptional and translational dynamics of DENV2- and ZIKV-infected human neural progenitor cells were analyzed. However, they did not investigate higher-order gene co-expression patterns or identify regulatory hub genes within infection-associated modules. Our study represents the first application of weighted gene co-expression network analysis (WGCNA) to these publicly available RNA-seq data from NCBI, allowing us to uncover network-level host responses and hub genes specific to DENV2 and ZIKV infection in neural progenitor cells. This statement is now clearly given in the last part of the introduction part.

---

## [Decision Letter · Decision Letter 1]

5 Apr 2026

Comparative Gene Co-Expression Network Analysis of Antiviral and Immune Responses to Dengue-2 and Zika Virus Infection in Human Neural Progenitor Cells ((hNPCs)

PONE-D-25-55299R1

Dear Dr. Siam,

We’re pleased to inform you that your manuscript has been judged scientifically suitable for publication and will be formally accepted for publication once it meets all outstanding technical requirements.

Kind regards,

Qiang Shawn Chen, Ph.D.

Academic Editor

PLOS One

Additional Editor Comments (optional):

Reviewers' comments:

Reviewer's Responses to Questions

**Comments to the Author**

1. If the authors have adequately addressed your comments raised in a previous round of review and you feel that this manuscript is now acceptable for publication, you may indicate that here to bypass the “Comments to the Author” section, enter your conflict of interest statement in the “Confidential to Editor” section, and submit your "Accept" recommendation.

Reviewer #1: All comments have been addressed

Reviewer #2: All comments have been addressed

2. Is the manuscript technically sound, and do the data support the conclusions?

Reviewer #1: Yes

Reviewer #2: Yes

3. Has the statistical analysis been performed appropriately and rigorously? 

Reviewer #1: Yes

Reviewer #2: Yes

4. Have the authors made all data underlying the findings in their manuscript fully available?

Reviewer #1: Yes

Reviewer #2: Yes

5. Is the manuscript presented in an intelligible fashion and written in standard English?

Reviewer #1: Yes

Reviewer #2: Yes

6. Review Comments to the Author

Reviewer #1: None.XXXXXXXXXXXXXXXXXXXXXXXXXXXXXXXXXXXXXXXXXXXXXXXXXXXXXXXXXXXXXXXXXXXXXXXXXXXXXXXXXXXXXXXXXXXXXX.

Reviewer #2: (No Response)

7. PLOS authors have the option to publish the peer review history of their article (what does this mean? ). If published, this will include your full peer review and any attached files.

**Do you want your identity to be public for this peer review?** For information about this choice, including consent withdrawal, please see our Privacy Policy .

Reviewer #1: No

Reviewer #2: **Yes:** Alan Goodman

---

## [Editor Report · Acceptance letter]

PONE-D-25-55299R1

PLOS One

Dear Dr. Siam,

I'm pleased to inform you that your manuscript has been deemed suitable for publication in PLOS One. Congratulations! Your manuscript is now being handed over to our production team.

Kind regards,

on behalf of

Dr. Qiang Shawn Chen

Academic Editor

PLOS One